# Trends in non-pharmaceutical intervention (NPI) related community practice for the prevention of COVID-19 in Addis Ababa, Ethiopia

**Damen Hailemariam, Abera Kumie** **\*, Samson Wakuma, Yifoker Tefera, Teferi Abegaz, Worku Tefera, Wondimu Ayele, Mulugeta Tamire, Shibabaw Yirsaw**

School of Public Health, College of Health Sciences, Addis Ababa University, Addis Ababa, Ethiopia

\* aberakumie2@yahoo.com

**Data Availability Statement:** All relevant data are within the manuscript and its Supporting Information files.

## Abstract

### Background

The COVID-19 pandemic has affected Ethiopia since March 13, 2020, when the first case was detected in Addis Ababa. Since then, the incidence of cases has continued to increase day by day. As a result, the health sector has recommended universal preventive measures to be practiced by the public. However, studies on adherence to these preventive measures are limited.

### Objective

To monitor the status of preventive practices of the population related to hand washing, physical distancing, and respiratory hygiene practices at selected sites within the city of Addis Ababa.

### Methods

Weekly cross-sectional non-participatory observations were done during the period of April-June, 2020. Data was collected using the Open Data Kit (ODK) tool in ten public sites involving eight public facilities targeted for individual observations. Ten individuals were randomly observed at each facility over two days a week at peak hours of public services. WHO operational definitions of the preventive behaviors were adopted for this study. Observations were conducted anonymously at gates or entrances of public facilities and places.

### Results

A total of 12,056 individual observations with 53% males and 82% in an estimated age range of 18–50 years age group were involved in this study. There was an increase in the practice of respiratory hygiene from 14% in week one to 77% in week 10, while those of hand hygiene and physical distancing changed little over the weeks from their baseline of 24% and 34%, respectively. Overall, respiratory hygiene demonstrated an increased rate of 6% per week, while hand hygiene and physical distancing had less than a 1% change per

**Funding:** The author(s) received no specific funding for this work.

**Competing interests:** The authors have declared that no competing interests exist.

week, Females and the estimated age group of 18–50 years had practice changes in respiratory hygiene with no difference in hand hygiene and physical distancing practices. Respiratory hygiene took about six weeks to reach a level of 77% from its baseline of 24%, making an increment of about 9% per week.

## Conclusion

The public practice of respiratory hygiene improved threefold whereas hand hygiene and physical distancing revealed no change. Regularly sustained public mobilization and mass education are required to sustain the achievements gained in respiratory hygiene and further hand hygiene and physical distancing.

## Introduction

The COVID-19 pandemic was originally described in Wuhan, China, in Dec 2019 [1]. Since then, it has spread to Europe, the USA, and later to Africa. The infection's nature of transmission is considered to be through droplets, and contact transmission [2]. The possibility of transmission through faeco-oral and airborne routes is still under investigation [3, 4]. Presymptomatic cases facilitate some of the transmission for a household clustering of infection and during travels [5–7]. Symptomatic infections appear to have the highest transmission potential compared to asymptomatic and pre-symptomatic when considering the transmission index [8]. The mortality pattern is devastating among elder age groups because of limited body immunity [9, 10].

Infection prevention and control (IPC) measures for emerging infectious diseases, like COVID-19, in health care settings can be challenging, especially in outbreaks where resources are limited [11]. Yet such measures, including early identification, prompt isolation, proper patient placement, adequate space ventilation, and proper use of personal protective equipment, are important in preventing and controlling the transmission of COVID-19 [12]. In addition, community prevention activities are considered the best strategies to reduce the healthcare burden, as they help slow transmission of the virus in the general population [13]. Community prevention strategies are a set of actions that persons and communities can implement to slow the transmission of COVID-19 by practicing hand hygiene, respiratory hygiene, and social distancing [14].

The Ministry of Health of Ethiopia provides daily updates on the number of new infections and reported mortalities associated with COVID-19 (https://www.worldometers.info/coronavirus/country/ethiopia). The first case in Ethiopia was declared on March 13, 2020, and the number of cases grew very slowly until the end of the first week of May 2020 (from 1 case on March 13 to 194 cases on 08 May 2020) [15]. However, the number of reported cases increased since then, reaching 110,074 cases and 1,706 deaths on 01 December 2020 (the time of the write-up of this manuscript) [16].

In Ethiopia, the Federal Ministry of Health (FMOH) and the Ethiopian Public Health Institute have created awareness about the virus transmission and prevention methods using different approaches, including mass media such as national television and radio during the partial-lock down period of five months since April, 20/2021. The interventions taken by the government ranged from banning public gatherings (over four persons) to limiting the number of passengers on public transport. In addition, all public service rendering institutions (such as food establishments, marketplaces, banks, and health facilities) provided hand-washing

facilities and maintained a physical distance of at least one meter between their clients. Furthermore, security forces and public authorities have shown direct involvement in maintaining mask use and maintaining physical distancing in public places such as marketplaces, public transport stations and terminals, and streets. Land borders were shut in an effort to control the entrance of the virus from abroad. These interventions have been associated with the decline of COVID-19 transmission in other countries [17] and that present the only effective methods in the absence of vaccines and medications. Data on monitoring the status of these interventions, however, is lacking in Ethiopia.

Our current study generates evidence on COVID-19 prevention practices of the residents of Addis Ababa (focusing on hand hygiene, physical distancing, and respiratory hygiene). The findings are expected to inform policy to sustain the related behaviors in the general population in our country.

## Methods

### Study design, study area, setting, and period

A cross-sectional observational design was conducted weekly between during April 24-June 28, 2020. Addis Ababa is the capital city of Ethiopia and is headquarters for the African Union and the Economic Commission for Africa. The city had a population of 5 million in Dec 2021 [18] and is sub-divided into ten administrative sub-cities. Sub-cities are divided into districts ("Wereda"). One of the biggest markets in Africa ("Merkato") is located in the heart of the city. There are about 131 Orthodox churches as well as several mosques. About 67% of the total vehicles (estimated to be about 800,000) are registered within Addis Ababa. There are several public and private banks in the city–including the Commercial Bank of Ethiopia (a state enterprise) that alone has over 120 branches in Addis Ababa.

This study was initiated as part of the Emergency Operation Center (EOC) preventive behavior monitoring activity at the College of Health Sciences (CHS), Addis Ababa University. CHS owns a teaching hospital that mandated an epidemic preparedness and management plan for its staff and visiting patients. The monitoring aimed to explore community practice levels regarding the three non-pharmaceutical interventions (NPI) for COVID-19 prevention including hand hygiene, physical distancing, and respiratory hygiene, in the ten administrative units (sub-cities) of Addis Ababa. The study was conducted by observing individual's behavior at selected sites when getting public services. These sites were selected based on the level of crowding and population mobility, and thus considered an indication of the risk for COVID-19 transmission. Facility sites by GPS are indicated in Fig 1.

**Study population.** The source population for the study was all individuals visiting the selected sites during the day of data collection. Study participants were individuals who visited the selected sites during the time of data collection.

**Monitoring protocol.** Before the data collection, a monitoring protocol had been prepared and approved by the research committee which was a standard operative procedure (SOP) used for the application of data collection. The protocol described the observation site selection, identified the list of public facilities to be observed, operationally defined the NPI practices, and the time and the procedure for the observation. First, "community center-public sites' were identified in each sub-city. These sites are characterized by having increased population mobility relative to the nearby neighborhoods. Then, one of the relevant sites was identified as a nucleus for the subsequent selection of government and public institutions and facilities for monitoring. Eight facility-sites were purposely selected based on selection criteria that included active public service provision, availability of adequate flow of service users, and appropriateness of the location for observation relative to increased population mobility.

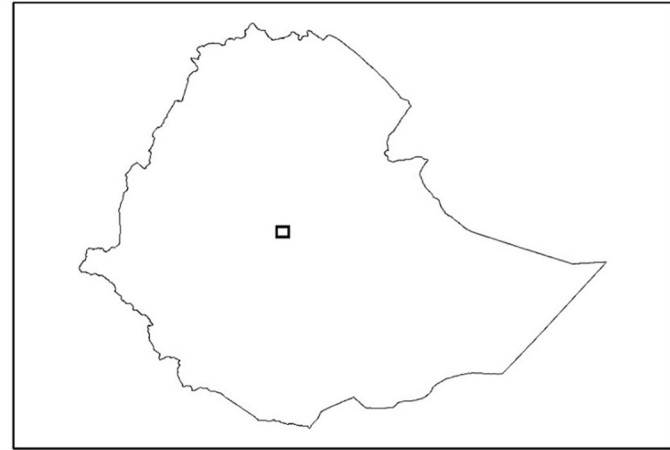
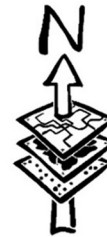
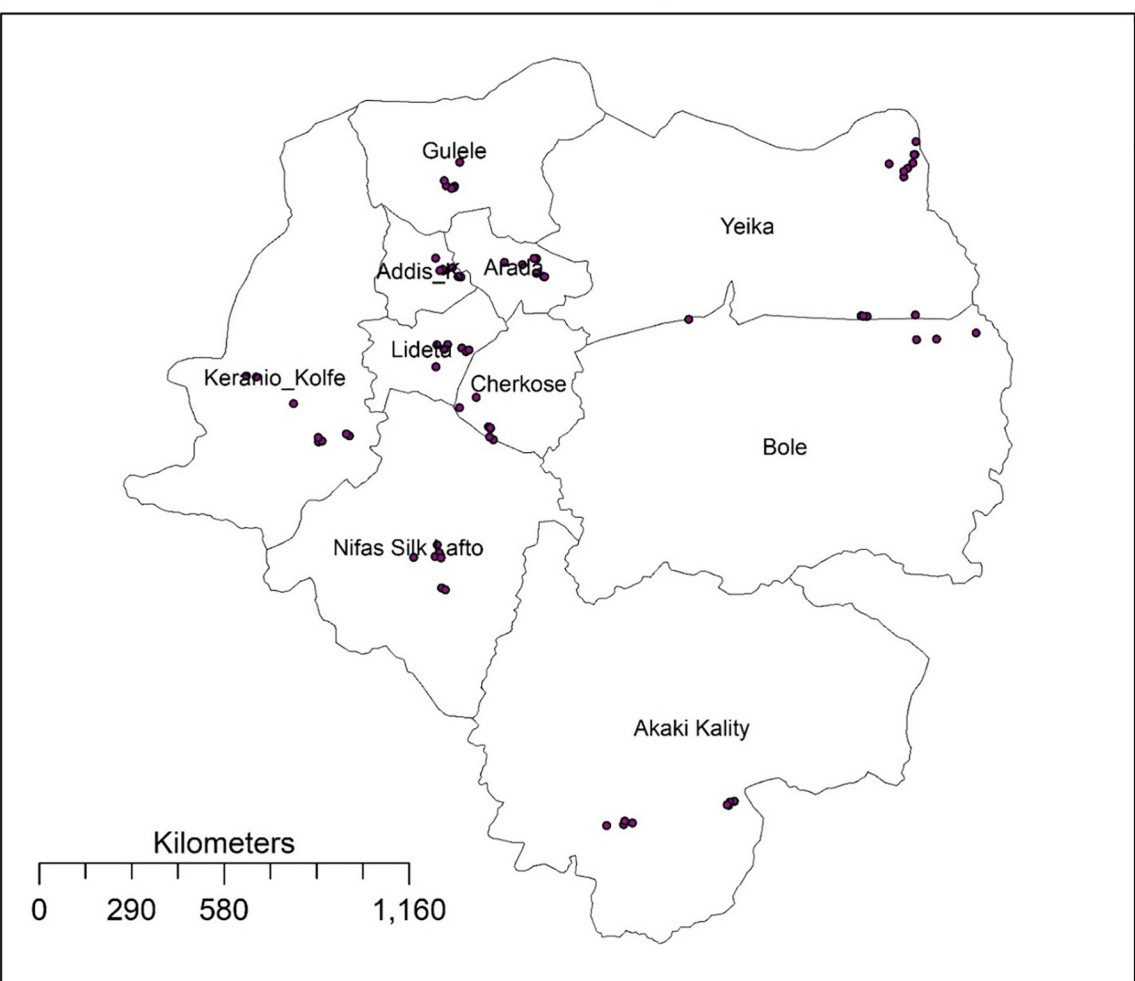

**Fig 1. Observational data collection sites in Addis Ababa.**

These sites represented religious places, health facilities, market places, banks, public transport, food and drink establishments, street crossing sites, and workplaces. Each public facility had ten observations.

We followed the protocol's provision in standardizing the context of observation. Fixing the distance of fewer than 10 meters from the observation point represented by the gate or entrance to the public services (buildings or open spaces like a traditional market) was maintained. This represents the distance that maintains the proximity of observation by the research assistant. The observation of an individual seeking a public service was designed anonymously to get valid behaviors of an individual's practice in reference to hand hygiene, physical distance, and respiratory hygiene. Ethical approval was obtained for this purpose. The research assistant found a convenient place that allowed his/her visual field to be optimum for an observation. The observer used a mobile phone ODK to record data without being identified by any person. This helps to distract suspecting his work, hence maintaining neutrality and data validity. The observer kept moving within the proximity of the gate in case there might be any suspicion of being identified. The research assistant was instructed to spend about 15–30 minutes on one site, which prevented identification. Finally, only two days in a week, weekdays (Wednesdays) and weekends (Sundays) were used for data collection to have weighted data represented by working days and holidays. The provisions of the context of monitoring are indicated in the S1 File.

**Data collection tools.** We prepared a data collection tool that had two sub-sections (attached in a S2 File). Section 1 addressed the location characteristics, while section 2 recorded the practice of the individual. The location checklist addressed information regarding the time of observation, the Global Positioning System (GPS) of each service facility location, the presence of any enforcement, and the availability of handwashing facilities, including water and detergents. The NPI practice section contained sex, estimated age, NPI practice in three categories (proper, improper, and none). Each NPI practice was operationally defined within the monitoring protocol following the WHO recommendation [19]. Key variable definitions are found in the S2 File.

**Data collection procedure.** One trained research assistant was assigned per sub-city, and one supervisor was appointed for two data collectors. The supervisor communicated with the field research assistant on each day of data collection to discuss the context and challenges of data collection. The field data were collected using the Open Data Kit tool (ODK) at the peak hours when the number of people was assumed to be highest at each observation site as stated in the protocol. The recorded ODK data were uploaded daily to the central server.

## Data management and analysis

Data from the main server were downloaded into Excel (Microsoft Corp.) and transferred into SPSS V 23 for data merging, cleaning, and analysis. Data were presented using descriptive statistics by calculating the proportion of practices for each week. The trend of the practice was evaluated using weekly differences in proportions (a current week with the previous week). The long-term trend was evaluated by visualizing the line graph and looking at the overall average of weekly differences. Limited statistical tests were done to see the significance of the overall practice with sex and the estimated age group. The weekly difference in the three practices was checked using a non-parametric Kruskal Wallis test. The average of differences between each week was considered to estimate the strength of the overall weekly change. Tables, line graphs, and charts were used to present data. The data set is in the S3 File.

**Ethical considerations.** The study received ethical approval from the Institutional Review Board (IRB) of the College of Health Sciences at Addis Ababa University. We had permission from IRB not to have consent as the results of the observations were meant to benefit the public at large and we did not seek any identifying information or photos of any individuals. This is indicated in the S4 File. The survey used a random observation tool that does not override

the rights of any individual or institution, and the observation was done purely anonymously. A weekly report on the progress of monitoring was submitted to the Ministry of Health of Ethiopia to help to take proactive actions.

## Results

### Study participants

A total of 12,056 people were observed from April 20 to June 28, 2020. More than half of these (59.2%) were observed on the weekdays (Table 1). We had 6,383 (52.9%) males and 5,673 (47.1%) females in the study. The majority, 9832 (81.6%), were in the estimated age group of 18–50 years.

### The overall trend of NPI community practice

Surgical masks predominated, (51%) relative to cloth masks, (41%), N95, (4%), and scarf, (4%). Proper hand hygiene in the community slowly increased from the baseline of 24.2% in week one to 32.5% in week three and then slowly declined to 23.7% in week 10. Similarly, proper physical distancing in the community increased from 34.4% in week one to 42.6% in week four and then slowly declined to 35.2% in week eight. On the contrary, proper respiratory hygiene in the community continuously increased from 14% in week one to 77% in week ten.

Driver's respiratory hygiene continuously increased from 13.8% in week one to 84.6% in week 10. Driver's assistants had 0% respiratory hygiene practice in week one but increased to 70.5% in week six, then slowly declined to 54.1% in week 10 (Table 2). The results also showed that adherence to the required vehicle occupancy less than or equal to 50% occupancy rate in commercial vehicles declined from 88.8% in week one to 68.8% in week 10. There was a sharp rise in the practice of respiratory hygiene, while those of hand hygiene and physical distancing did not change or declined during the last week of observation (Fig 2). Week 4 was the turning point to observe an increase, while the highest mask used in a population took seven weeks. The trend in hand hygiene and physical distancing through the ten weeks was stable relative to respiratory hygiene.

The three combined hand and respiratory hygiene practices with physical distancing managed by an individual were below 10% in any week. It increased slowly from 4.7% in week one to 10.8% in week four and then declined to 8.1% in week 10.

**Table 1. Number of people observed (April 20– June 14, 2020, Addis Ababa).**

| Observation week | Weekdays | Weekend | Total observations # (%) |
|---|---|---|---|
| | observations # (%) | observations # (%) | |
| Week 1 (Apr20-26) | 225 (26.3) | 629 (73.7) | 854 (7.1) |
| Week 2 (Apr27-May3) | 723 (62.1) | 442 (37.9) | 1,165 (9.7) |
| Week 3 (May 4-May 10) | 804 (63.6) | 460 (36.4) | 1264 (10.5) |
| Week 4 (May 11-May 17) | 719 (60.0) | 480 (40.0) | 1,199 (9.9) |
| Week 5 (May 20- May 24) | 775 (62.2) | 470 (37.8) | 1,245 (10.3) |
| Week 6 (May 27-May 31) | 706 (59.1) | 488 (40.9) | 1,194 (9.9) |
| Week 7 (June 3-June 7) | 799(62.0) | 489 (38.0) | 1,288 (10.7) |
| Week 8 (June 10-June 14) | 788(61.7) | 490 (38.3) | 1,278 (10.6) |
| Week 9 (June 15- June 21) | 802 (62.6) | 479 (37.4) | 1,281 (10.6) |
| Week 10 (June 22- June 28) | 798 (62.0) | 490 (38.0) | 1,288 (10.7) |
| **Total** | **7,139 (59.2)** | **4,917 (40.8)** | **12,056 (100)** |

## Varation in practices

Overall, we have found statistical differences for the three practices (p<0.05), and within-subject differences by sex, age, and day of observation (P<0.05). Females and age category of 18–50 years had statistical differences in the practice for respiratory hygiene. The practices observed in the week-days had an increased proportion over the week-end days (p<0.05) for hand and respiratory hygiene.

## The trend of respiratory hygiene by observation facilities

There was a similar pattern of an increasing trend in all eight facilities with varying intensity of weekly increases. Banks and health facilities showed fast and stable growth rates over the ten

**Table 2. Weekly pattern of a community practice of non-pharmaceutical interventions (Addis Ababa, 2020).**

| NPI | NPI Practices | Week 1 (Apr 20–26) | Week 2 (Apr 27-May3) | Week 3 (May 4–10) | Week 4 (May 11–17) | Week 5 (May 20–24) | Week 6 (May 27–31) | Week 7 (June 3–7) | Week 8 (June 10–14) | Week 9 (June 15–21) | Week 10 (June 22–28) |
|---|---|---|---|---|---|---|---|---|---|---|---|
| Hand hygiene | **Proper hand hygiene** | **178 (24.2)** | **328 (33.1)** | **350 (32.5)** | **323 (32.0)** | **305 (28.8)** | **26 1 (25.9)** | **288 (26.1)** | **290 (26.8)** | **281 (28.5)** | **254 (23.7)** |
| | Improper hand hygiene | 88 (11.9) | 93 (9.4) | 101 (9.4) | 76 (7.5) | 78 (7.4) | 80 (7.9) | 121 (10.9) | 105 (9.7) | 123 (12.4) | 103 (9.6) |
| | No hand hygiene | 469 (63.8) | 571 (57.6) | 625 (58.1) | 610 (60.4) | 674 (63.8) | 667 (66.2) | 692 (62.8) | 684 (63.4) | 582 (59.1) | 715 (66.7) |
| | **Total** | **735** | **992** | **1076** | **1009** | **1057** | **1008** | **1101** | **1079** | **986** | **1072** |
| Physical distance | **Proper physical distance** | **294 (34.4)** | **462 (39.7)** | **491 (38.8)** | **511 (42.6)** | **46 7 (37.5)** | **444 (37.2)** | **479 (37.2)** | **496 (38.8)** | **503 (39.3)** | **454 (35.2)** |
| | Improper physical distance | 408 (47.8) | 531 (45.6) | 593 (46.9) | 551 (46.0) | 655 (52.6) | 643 (53.9) | 699 (54.3) | 662 (51.8) | 711 (55.5) | 710(55.1) |
| | No physical distance | 152 (17.8) | 172 (14.8) | 180 (14.2) | 137 (11.4) | 123 (9.9) | 107 (9.0) | 110 (8.5) | 120 (9.4) | 67 (5.2) | 124(9.6) |
| | **Total** | **854** | **1165** | **1264** | **1199** | **1245** | **1194** | **1288** | **1278** | **1281** | **1288** |
| Respiratory hygiene | **Proper respiratory hygiene** | **205 (24.0)** | **301 (25.8)** | **405 (32.0)** | **575 (48.0)** | **746 (59.9)** | **845 (70.8)** | **985 (76.5)** | **980 (76.7)** | **999 (78.0)** | **995 (77.3)** |
| | Improper respiratory hygiene | 79 (9.3) | 174 (14.9) | 213 (16.9) | 271 (22.6) | 151 (12.1) | 153 (12.8) | 198 (15.4) | 214 (16.7) | 208 (16.2) | 201 (15.6) |
| | No respiratory hygiene | 570 (66.7) | 690 (59.2) | 646 (51.1) | 353 (29.4) | 348 (28.0) | 196 (16.4) | 105 (8.2) | 84 (6.6) | 74 (5.8) | 92 (7.1) |
| | **Total** | **854** | **1165** | **1264** | **1199** | **1245** | **1194** | **1288** | **1278** | **1281** | **1288** |
| Drivers' respiratory hygiene | **Proper respiratory hygiene** | **11 (13.8)** | **34 (23.0)** | **91 (55.2)** | **133 (78.7)** | **159 (84.6)** | **170 (89.5)** | **167 (84.8)** | **165 (82.9)** | **156 (82.5)** | **168 (84.6)** |
| | Improper respiratory hygiene | 1 (1.3) | 11 (7.4) | 15 (9.1) | 11 (6.5) | 23 (12.2) | 17 (8.9) | 29 (14.7) | 30 (15.1) | 30 (15.9) | 27 (13.6) |
| | No respiratory hygiene | 68 (85.0) | 103 (69.6) | 59 (35.8) | 26 (15.3) | 6 (3.2) | 3 (1.6) | 1 (0.5) | 4 (2.0) | 3 (1.6) | 4 (2.0) |
| | **Total** | **80** | **148** | **165** | **170** | **188** | **190** | **197** | **199** | **189** | **199** |
| Driver assistants' respiratory hygiene | **Proper respiratory hygiene** | **0%** | **29 (19.6)** | **60 (37.3)** | **87 (52.1)** | **119 (62.6)** | **141 (70.5)** | **135 (69.9)** | **118 (60.5)** | **111 (59.0)** | **106 (54.1)** |
| | Improper respiratory hygiene | 3(4.0) | 19(12.8) | 29 (18.0) | 44 (26.3) | 61 (32.1) | 48 (24.0) | 57 (29.5) | 72 (36.9) | 73 (38.8) | 80 (40.8) |
| | No respiratory hygiene | 72 (96.0) | 100 (67.6) | 72 (44.7) | 36 (21.6) | 10 (5.3) | 11 (5.5) | 1 (0.5) | 5 (2.6) | 4 (2.1) | 10 (5.1) |
| | **Total** | **75** | **148** | **161** | **167** | **190** | **200** | **193** | **195** | **188** | **196** |
| Vehicle occupant capacity | **Less than or equal to 50% capacity** | **71 (88.8)** | **122 (83.0)** | **114 (69.1)** | **110 (65.1)** | **128 (68.1)** | **128 (67.4)** | **136 (69.0)** | **140 (70.4)** | **132 (69.8)** | **137 (68.8)** |
| | Greater than 50% capacity | 9 (11.3) | 25 (17.0) | 51 (30.9) | 59 (34.9) | 60 (31.9) | 62 (32.6) | 61 (31.0) | 59 (29.6) | 57 (30.2) | 62 (31.2) |
| | **Total** | **80** | **147** | **165** | **169** | **188** | **190** | **197** | **199** | **189** | **199** |

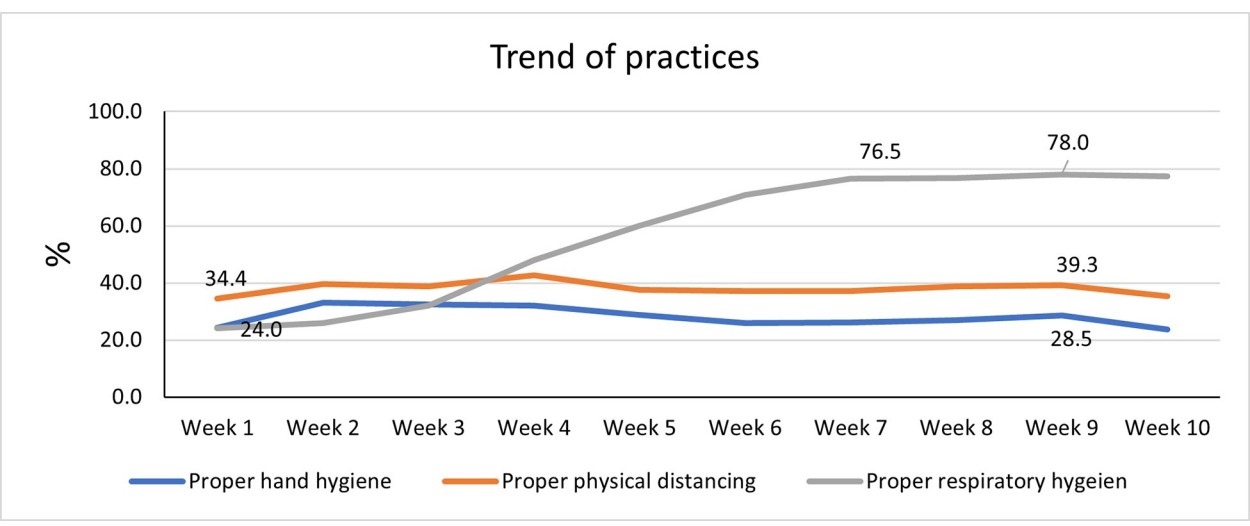

**Fig 2. The overall trend of the proper community practices for the three NPIs.**

weeks, from 30% in week one to 90% in week ten and from 30% in week 1 to 86% in week 10, respectively (Fig 3).

## Discussion

Non-pharmaceutical interventions are the primary preventions to restrict the spread of COVID-19 in the setting of limited access to vaccines. Overall, our results demonstrated that the use of masks was stable at around 80%, while hand hygiene and physical distancing were not observed to go beyond 40%. The optimum application of the three practices was limited in

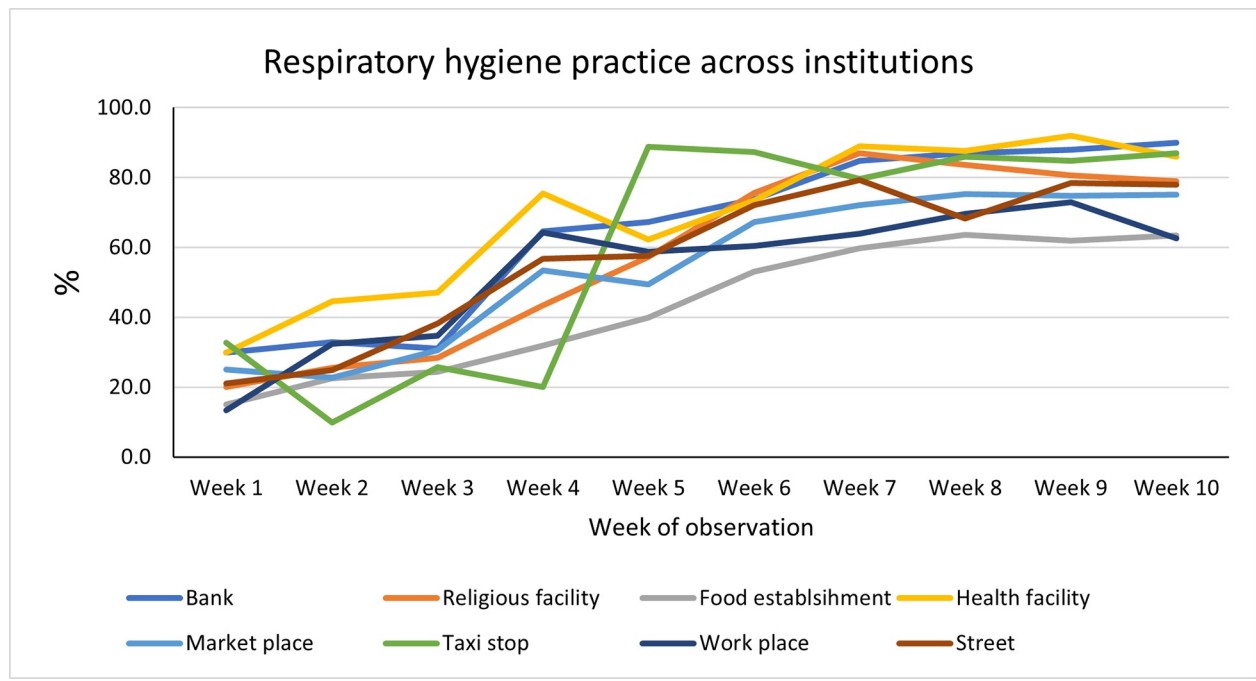

**Fig 3. The trend of the proper respiratory hygiene practices by public facilities.**

our study and only attained for respiratory hygiene. The World Health Organization greatly acknowledged the combined practice of non-pharmaceutical interventions as a frontline tool to combat the infection [19, 20]. The non-pharmaceutical interventions were effective in containing the outbreak demonstrated by the number of cases that could have been 67-fold higher at the time without these interventions [21]. Systematic reviews have shown the frequent practice of hand hygiene, the use of physical distancing of 1 m or more, and mask utilization have a potential benefit of lowering the COVID-19 transmission [22, 23]. These interventions are effective with optimum compliance if not universal in a community as has been strongly proposed in China for a high-risk population [24] The Nigerian Medical Association strongly criticized the limited use of non-pharmaceutical prevention protocols by citizens [25]. The low level of mask using was mainly attributed to risky social behaviors, perceived risk of transmission, gender and age differences [26]. This study also showed the variation of the mask use over time across many countries: some having gradually increased, others showing a declining trend. The community's access to affordable masks in our study should have provided a sense of protection compared to hand hygiene and physical distancing. The use of cloth masks that are easily accessible, cheap, and existing in many different options has provided a degree of protection.

The present study has shown major improvements in the practice of respiratory hygiene compared to hand hygiene and physical distance during the period of our observation. The improvement started to show significant change after week five matched with when security forces started to enforce primary preventions in response to the state of emergency related to the epidemic crisis in Ethiopia. Although the state of emergency was declared on 20 April 2020, the full breadth of the impact came later, several weeks later of massive promotional provisions targeting the public at large and public service rendering institutions in particular. At the time of the study, the State Emergency in Ethiopia was characterized to have partial lockdown of public places, including the closing of educational facilities and workplaces coupled with strict use of masks at public places, such as transport and public service places, which resulted in a rapid increase during our monitoring period. According to the state of emergency provisions, public mobility was not restricted to seeking food, transport, health care, and marketing services. There are reports from other countries that NPI interventions during lockdown periods indicating them as effective tools to interrupt COVID-19 transmission [27, 28], although the benefit of physical distancing and masking has been uncertain due to lack of strong study designs [28]. The state of emergency mandated private and government service providers to avail handwashing facilities with detergents for their clients who seek services. Handwashing facilities were usually placed at the gate or entrance of the institution to encourage hand hygiene before accessing the service. The perception of self-protections by using masks might have contributed towards the increasing practice. On the contrary, a small portion of the public, about 20%, did not use a mask that might be linked to the negative attitudes towards mask ineffectiveness to reduce the transmission [29].

Despite the public enforcement, the practice of proper hand hygiene did not show much change over the ten weeks of observation and did not go beyond 30%. Two reasons were suggested from our observation. Firstly, the media company and involvement of the enforcing volunteers had massive promotions in the first four weeks during our data collection period around churches, transport stations, and other public places. This may have contributed to a change in handwashing practice in those weeks but not much beyond these weeks. On the other hand, the unavailability of water and soap could not be sustained in many facilities, except institutions such as banks and health facilities. Generally, the provision of soap was initially challenging, which was followed by a shortage and total disappearance of water by many public service providers. We believe these challenges happened at the same time of weak

enforcement when security and other partners discontinued the initiated interventions. There was a shift from washing hands to using hand sanitizer by some members of the population, especially the youth. Our observations had limited coverage of the youth; thus, we might have underestimated the proportion of handwashing in a community. The youths had limited needs for seeking services in facilities that were involved in our study.

The maintenance of physical distancing at a minimum of one meter between any two individuals had a similar trend with handwashing practice. However, the former had an overall performance of below 40% relative to the 30% of the latter practices. Physical distancing was challenging for social services, predominantly observed in open markets, walking and crossing on a street, and seeking public transport sectors. Individuals in an open market traditionally tend to cluster around a seller to make transactions. Seeking public transport services at peak hours makes service seekers get closer around the gate of transport vehicles. The duration of exposure and physical distancing has been shown to be an important predictor of the transmission of SARS-CoV-1 [30]. A systematic review of 172 studies reported that physical distancing of one meter was associated with a large reduction in the number of new infections in different countries [31]. Taking the economic status of Ethiopia, the lack of change in the physical distancing at the public level in this study was a missed opportunity to avert the potential occurrence of new cases in the city. Hand hygiene and physical distancing are effective, low-cost interventions that could reduce the number of new cases [17, 31]. We believe limited adherence in hand hygiene and physical distancing are missed opportunities to avert new infections in a community.

Ethiopia possesses traditions that encourage the community to get together, such as burial events, traditional religious festivals, attending churches, and traditional open markets. The State emergency was not universal in the restriction of mass mobility during its active implementation phase, which might be a factor for the spread of the infection. Clients were advised to perform the three practices strictly during the initial wave of the epidemic. The enforcement of NPIs became weaker, however, a week after lifting off the state emergency. There were incidences during the socio-cultural events when physical distancing was ignored because of the population's crowding, such as religious holidays. Contrary with this finding, a study from China reported that almost all respondents avoided going to public places and reduced celebrating spring festivals due to the COVID-19 epidemic [32]. The possible reasons for the difference in adherence might be associated with the socio-cultural contexts in a given country and the degree of enforcing preventive practices.

The extent and trend of mask utilization had visible differences compared to physical distancing and hand hygiene. Mask utilization was observed to have an increasing trend until it stabilized at about 80% towards the end of our study time. The continued security enforcement, mass media mobilization, the restrictions of public services without a mask, and the provisions of regular information about COVID-19 using national TV and radio were the main contributing factors. The multiple sources of information and public enforcement were believed to enhance the attitude of service seekers, which greatly modified their behavior towards the use of a mask. We cannot deny that access to multiple types of masks, such as surgical, N95, and cloth masks and their local production at affordable prices has critically sustained the trend of mask utilization. The local production of face masks in low-middle income countries has been indicated as a strategy to contain the epidemic [33]. Before or after the state of emergency, the time of the observation and the age of study subjects could make a difference in the proportion of mask using. There was a preference of using masks among the young population in Poland, 60%, in a study undertaken before the government's enforcement [34, 35], which is relatively lower than ours. Our study was undertaken after the declaration of State Emergency, which has brought multiple interventions to impact the behaviors of individuals

towards using a mask. Overall, the universal use of masks in public has been considered as a means to reduce the transmission of respiratory viruses COVID-19 in previous studies [36, 37].

We disaggregated the mask by type of facilities to get insights about the factors affecting mask using. The trend of respiratory hygiene practice visibly increased among service providers involving health care facilities, banks, and transport services compared with other facilities included in this study. There were assigned controllers and guards checking for the use of face-masks and maintenance of physical distancing and hand hygiene, who did not allow individuals to get a public service without these practices.

The relatively improved practices at bank and health facilities may have been related to the fact that these institutions have committed the resources to provide handwashing facilities with detergents and that their clients would have a relatively higher level of awareness and favorable attitude towards the preventive practices As there are variabilities in people's beliefs about the effectiveness as well as the rates of engagement in the use of non-pharmaceutical interventions, public health efforts should focus on increasing perceived severity and threats of the COVID-19 while promoting the preventive measures as effective tools [37, 38] and enforcing the use of them as necessary. On the contrary, the adherence to mask using and physical distancing in open markets and food establishments will likely continue to exist as a challenge. Food and drink establishments require close attention because people eat in groups, and physical distancing was never practically applied. Onsite dining with indoor and outdoor seats was identified as a risky situation by the CDC [13].

A strength of our study was observing individuals randomly and anonymously at times of service provisions during peak hours in key social providing services. This approach would have reduced respondent's bias. The provision of weekly information to the Ministry of Health was an important input to enhance informed decisions to the public. We assumed observing individuals at the time of possible transmission in the presence of increased mobility and crowding provided better insights into the possibility of the transmission potential. We recognize the limitations of our study. We could not determine the presence of interview bias that could happen over time, although we had regular supervisions and meetings with research assistants. The data collected was for a limited time observation and may not represent the profile individual's behavior on a given day. We recognize the accuracy and reproducibility of age categories used are only estimates and variance was likely in this regard. Finally, our study cannot predict the trends in community practice across the whole country in the absence of state wide interventions.

## Conclusions and recommendations

The overall community prevention practices for the COVID-19 pandemic were very low in Addis Ababa, Ethiopia, although respiratory hygiene singly showed a relatively increased rate. Despite the relatively low adherence of the community to proper handwashing and proper physical distancing, the success of proper respiratory hygiene was encouraging. Strengthening social mobilization using available media and consistent enforcement mechanisms to get sustained performances in community practices would be of value.

## Supporting information

**S1 File. Study protocol.**
(DOCX)

**S2 File. Data collection checklist.**
(DOCX)

**S3 File. Data2 set SPSS.**
(SAV)

**S4 File. IRB approval letter_08June2020.**
(PDF)

## Acknowledgments

The authors are grateful to the College of Health Sciences and the School of Public Health of Addis Ababa University for allowing this study to happen. The leadership of Dr Dawit Wendimagegn, who is the Chief Executive Director of the College of Health Sciences is greatly thanked for his support to undertake this study.

## Author Contributions

**Conceptualization:** Damen Hailemariam, Abera Kumie, Samson Wakuma, Yifoker Tefera.

**Data curation:** Abera Kumie, Yifoker Tefera, Wondimu Ayele, Mulugeta Tamire, Shibabaw Yirsaw.

**Formal analysis:** Abera Kumie, Samson Wakuma.

**Methodology:** Abera Kumie, Samson Wakuma, Yifoker Tefera, Wondimu Ayele, Mulugeta Tamire.

**Software:** Wondimu Ayele.

**Supervision:** Damen Hailemariam, Abera Kumie, Teferi Abegaz, Worku Tefera.

**Validation:** Abera Kumie, Teferi Abegaz.

**Visualization:** Damen Hailemariam, Wondimu Ayele.

**Writing – original draft:** Damen Hailemariam, Abera Kumie, Teferi Abegaz, Worku Tefera, Wondimu Ayele, Mulugeta Tamire.

**Writing – review & editing:** Damen Hailemariam, Abera Kumie, Samson Wakuma, Yifoker Tefera, Teferi Abegaz, Worku Tefera, Wondimu Ayele, Mulugeta Tamire.

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
