## [Decision Letter · Decision Letter 0]

19 Mar 2021

PONE-D-21-03031

Trends in non-pharmaceutical intervention (NPI) related community practice for the prevention of COVID-19 in Addis Ababa, Ethiopia

PLOS ONE

Dear Dr. Abera Kumie

Thank you for submitting your manuscript to PLOS ONE. After careful consideration, we feel that it has merit but does not fully meet PLOS ONE’s publication criteria as it currently stands. Therefore, we invite you to submit a revised version of the manuscript that addresses the points raised during the review process.

As the Academic Editor, I agree with the reviewers that the topic is an important one and the large number of evaluations  in the setting you describe help to capture the overall poor compliance with the NPIs under study and represents real world observations. I also agree with the reviewers that significant work is required to make this manuscript suitable for publication.

Please address all the reviewer comments with the exception of  Reviewer 2's comments about sample size given this is a descriptive epidemiologic study. You can look at significance of differences between groups and provide some mean and median values where appropriate . Both reviewers commented about the details surrounding the NPIs and would agree more details are required other than a general WHO reference and the respiratory hygiene which is presumably referring to masks wearing needs to be defined more precisely. The references to large amounts of transmission from asymptomatic cases needs  updating  (see  Koh WC, Naing L, Chaw L, Rosledzana MA, Alikhan MF, Jamaludin SA, Amin F, Omar A, Shazli A, Griffith M, Pastore R, Wong J. What do we know about SARS-CoV-2 transmission? A systematic review and meta-analysis of the secondary attack rate and associated risk factors. PLoS One. 2020 Oct 8;15(10):e0240205). You also need to refine what part of the WHO document to which you are referring in reference 14 and the Jefferson review listed as reference 23 has been updated to a recent new date from Nov 2020. The Discussion is too long and unfocused in areas and needs to be tightenmded and agree with the comments of Reviewer 1 in this context. The references are not in the standard format for the Journal and case, formatting, access dates and full urls must be provided.

You are strongly recommended to have someone whose native language is English review since there are many grammatical and syntax errors and it impacts negatively on the readability.

Please submit your revised manuscript  within 45 calendar days of the receipt of this letter.  If you will need more time than this to complete your revisions, please reply to this message or contact the journal office at plosone@plos.org. Please include the following items when submitting your revised manuscript:

We look forward to receiving your revised manuscript.

Kind regards,

John Conly, MD

Academic Editor

PLOS ONE

Journal Requirements:

2. Please provide additional details regarding participant consent. In the ethics statement in the Methods and online submission information, please ensure that you have specified (a) whether consent was informed and (b) what type you obtained (for instance, written or verbal, and if verbal, how it was documented and witnessed). If your study included minors, state whether you obtained consent from parents or guardians. If the need for consent was waived by the ethics committee, please include this information.

"The data collection was funded by the School of Public Health of Addis

Ababa University. The Authors have contributed professionally in view of supporting the

Ministry of Health in enhancing proactive informed decisions."

4. Please amend the manuscript submission data (via Edit Submission) to include author Abera Kumie.

5. Please include your tables as part of your main manuscript and remove the individual files. Please note that supplementary tables (should remain/ be uploaded) as separate "supporting information" files

Reviewers' comments:

Reviewer's Responses to Questions

**Comments to the Author**

1. Is the manuscript technically sound, and do the data support the conclusions?

Reviewer #1: Partly

Reviewer #2: Partly

2. Has the statistical analysis been performed appropriately and rigorously? 

Reviewer #1: No

Reviewer #2: No

3. Have the authors made all data underlying the findings in their manuscript fully available?

Reviewer #1: Yes

Reviewer #2: Yes

4. Is the manuscript presented in an intelligible fashion and written in standard English?

Reviewer #1: No

Reviewer #2: No

5. Review Comments to the Author

Reviewer #1: Overall, this is an important area of study given the ongoing COVID-19 pandemic. Public participation with non-pharmaceutical interventions (NPIs) including hand hygiene, respiratory hygiene and physical distancing are crucial to successful containment and prevention of ongoing transmission in the community. Observational studies such as this do help identify areas for improvement.

However, I feel this paper requires significant work before it should be considered for publication. My main concerns are; 1) the lack of statistical analysis with respect to the descriptive data presented and 2) need for language editing and clarity, particularly in the discussion.

ABSTSTRACT

Overall, well written. However, I find the number of observations described in the methods section somewhat confusing and misleading compared to how this is described in the methods section of the paper itself. I would suggest simplifying this description. The statistics presented in the results lack significance analysis (i.e. p-values). I do not think the conclusion in the abstract necessarily match the results described.

INTRODUCTION

Paragraph 1 – “Pre-asymptomatic cases facilitate most of the transmission” Do you mean pre-SYMPTOMATIC here?

Paragraph 2 – I think the first two sentences are not necessary for your background argument.

Paragraph 4 – I think the end of the introduction would benefit from a clearer study objective statement as a separate small paragraph.

METHODS

Paragraph 1 – This paragraph may work better as part of the introduction.

Paragraph 3 – I would clearly state that all age groups (including children) were included in the study population.

Paragraph 4 – It would be helpful to expand on which types of sites (and how many) were included for observation. I would also include more detail about how you observed drivers and public transportation as this was specifically mentioned in the results section.

Paragraph 5 – I would include a copy of the two check lists used in the study.

Paragraph 6 – A minimum of 30 minutes of observation time is suggested but was a maximum time period defined? Why was a standard observation time not used?

Paragraph 7 – With respect to the age variable, I am concerned about the accuracy and reproducibility of estimated age categories used in this study. What is the rationale for why these particular age groupings were used? I would also suggest clearly stating how you defined proper and improper respiratory hygiene, hand hygiene and physical distancing for the purpose of this study, rather than referencing WHO. It would also be important to clarify if both soap/water and hand sanitizers were considered adequate hand hygiene as there is some discussion about this later in the paper.

Paragraph 8 – Descriptive statistics do not appear to be used for analysis. This needs to be defined further in this section, the applied to the data and presented in the results/table section.

Paragraph 9 – With respect to consent, was the public or any of the places of business used as observation sites made aware of this study?

RESULTS, FIGURES AND TABLES

Paragraph 1/Table 1 – Week one appears to be an outlier with respect to number of observations made on a weekday vs weekend compared to the other weeks is there is reason for this? I don’t believe it was addressed in the discussion, either.

Table 2 is not necessary as this data is easily described in the text of paragraph 2.

Paragraph 3/Table 3 – No statistical analysis performed therefore unable to know if differences described are significant based on the number of observations made. Why was transportation presented separately in this table? Why not a breakdown by facility type as well (see my comments regarding figure 3)?

Paragraph 4/Figure 3 – This is interesting data regarding facility type but not particularly clear when presented in this format. I would suggest a breakdown and analysis similar to Table 3.

Paragraph 5 – I think it is important to show the compliance data with all three NPI measurements at the same time as it demonstrates how challenging it can be for the general public to adhere to multiple recommendations at the same time. However, I do not feel the discussion of the various other 2 out of 3 NPI combinations is particularly useful.

Table 4 – The idea of “risk” categories was not defined in the methods or mentioned in the results section. These categories are arbitrary and I do not feel this particular table/analysis should be included.

DISCUSSION AND CONCLUSION

The discussion and conclusions requires a significant amount of editing for clarity. There should also be separate paragraphs specifically addressing limitations of the study as well as specific future interventions or research.

In paragraph 1, use of face masks was considered satisfactory but what is your definition of this? There is no discussion around types of masks, observed donning/doffing procedures, re-using old/soiled masks. These issues may be beyond the scope of this particular study but should be mentioned in the limitations section.

Paragraph 2 suggests that declining compliance with NPIs in other communities is common over time but it would be helpful to have a discussion around why this occurs.

Paragraph 3 addresses how the declaration of a state of emergency and other measures, such as limiting people’s movements in the community, are also important interventions. This paragraph requires some editing to clarify this point and what the impact was specifically in Ethiopia.

Paragraph 4 – There are two competing ideas in this paragraph that need to be separated and explored further. The issue of available soap/water stations versus hand sanitizer is important. It is unclear if hand sanitizer was readily available or if it was included an appropriate method of sanitization in the methods of this study. The second issue is around public education campaigns, specifically around hand hygiene. I public education campaigns should be described and explored further with respect to their effect on this study’s results and include the other NPIs as well.

Paragraph 5 – The last sentence of this paragraph is not necessary and I would suggest removing it. Otherwise, this paragraph is very important. Are there any studies that have attempted to look at physical distancing strategies that could be useful in busy, open air market settings from other countries?

Paragraph 6 –Were religious and cultural leaders involved in promoting NPIs or encouraging celebrating religious days in smaller groups or at home?

Paragraph 7 – This paragraph seems out of place and many sentences are redundant with ideas addressed in other paragraphs.

Reviewer #2: Dear Authors:

Thank you for your effort in putting together this manuscript, covering a very important topic.

Unfortunately, I have several concerns, which will need to be adequately addressed prior to my endorsement of your paper for publication in PLOS ONE. I will outline them as follows:

1. PLOS ONE does not copyedit accepted manuscripts, so the language in submitted articles must be clear, grammatically correct, and unambiguous. Given the number of grammatical errors and confusingly worded statements in your paper, I would strongly suggest that you seek independent editorial assistance prior to submitting a revision. Very importantly, please make it clear that by the term respiratory hygiene, you are referring to masking (as the term respiratory hygiene is also used to describe adherence to cough etiquette).

2. Your data in the results section is simply a description of the percentage of different groups of people that you observed adhering to your there non-pharmaceutical interventions. From what I can tell, there is no described sample size nor is there any testing for statistical significance (with a standard alpha = 0.05 and power typically designated at 80%). Without an inclusion of any hypothesis testing and p-values, one could conceivably state that all of the differences you observed between (and within) different groups over time was due to chance alone. If necessary, you may want to avail of a statistician.

3. It would be helpful if you could describe in more detail the nature of the observations and the observers. Do you believe that there was an element of observer bias over time, and that people changed their behavior because they knew they were being watched in these different settings? Clearly, it appears that adherence to masking increased to a high degree in the context of security force related enforcement. However, why was similar enforcement not seen with physical distancing? (This may have been due to the challenges of enforcing distancing in large crowded areas). It also seems fairly clear that handwashing rates did not increase due, in large part, to a lack of available soap and water as you mention. However, you mention that hand sanitizer use, which is more common amongst youth, was not included in your results. Why is this the case?

4. With respect to your age demographic info (i.e < 18 yo, 18-50 yo and >50 yo), you mention that it was easy to categorize people into these groups based on appearance alone. Although it may be easy to tell a 35 year old apart from a 75 year old, it may have been difficult to tell a 17 year old apart from a 19 year old. Ultimately, these discrete categories seem a bit arbitrary, given that they are "guesstimates" at best.

6. PLOS authors have the option to publish the peer review history of their article (what does this mean?). If published, this will include your full peer review and any attached files.

Reviewer #1: No

Reviewer #2: No

---

## [Author Response · Author response to Decision Letter 0]

11 Jul 2021

Dear Reviewers

The reviewing questions and concerns were very helpful. We have all accommodated giving a point by point response to each concern in three separate files. The English language was heavily edited.

We are hoping to satisfy your concern.

<Many thanks for the understanding

---

## [Editor Report · Decision Letter 1]

9 Aug 2021

PONE-D-21-03031R1

Trends in non-pharmaceutical intervention (NPI) related community practice for the prevention of COVID-19 in Addis Ababa, Ethiopia

PLOS ONE

Dear Dr. 

Thank you for submitting your revised manuscript to PLOS ONE.  After careful consideration, we feel that it has merit but does not fully meet PLOS ONE’s publication criteria as it currently stands. Therefore, we invite you to submit a re-revised version of the manuscript that addresses the points raised during the review process.

You have addressed the majority of the peer reviewer and Academci Editor comments so thank you for your efforts in this regard. The study fulfills a key gap in evaluating "real world" COVID-19 mitigtation measures by the  general public in a large African city and hence are very meaningful. The contents are sound with respect to the study and its results. However the following points  must  be addressed.

1. The  Engish grammar and syntax are not at a standard which would be acceptable. I have taken the liberty of addressing the English grammatical errors throughout the manuscroipt and have placed the corrections in a series of strikethroughs and sticky notes. Please review carefully.and make the changes in the Word version and ensure no scientific content has been altered .

2. The references are in major need of "cleaning" to ensure they are in the appropriate format for the Journal with respect to abbreviations of the journal names, italics for Latin terms, case, punctuation,  etc and must be completely correct.

3. The supplemental files have spelling errors, syntax errors and English grammatical errors and need to be carefully reviewed and approriate corrections made. 

Please submit your revised manuscript within 30 calendar days of receipt of this letter . If you will need more time than this to complete your revisions, please reply to this message or contact the journal office at plosone@plos.org. Please include the following items when submitting your revised manuscript:

We look forward to receiving your revised manuscript.

Kind regards,

John Conly, MD

Academic Editor

PLOS ONE
---

## [Author Response · Author response to Decision Letter 1]

4 Sep 2021

We have all accomdated the comments

---

## [Editor Report · Decision Letter 2]

23 Sep 2021

PONE-D-21-03031R2

Trends in non-pharmaceutical intervention (NPI) related community practice for the prevention of COVID-19 in Addis Ababa, Ethiopia

PLOS ONE

Dear Dr. Tefera,

Thank you for submitting your revised manuscript to PLOS ONE. There remain several required revisions, mainly in the English grammar, font shifts and cleaning of  references .  Therefore, we invite you to submit a revised version of the manuscript that addresses the points raised during the review process.

Please see the attached pdf with the suggested changes. 

We look forward to receiving your revised manuscript.

Kind regards,

John Conly, MD

Academic Editor

PLOS ONE
---

## [Author Response · Author response to Decision Letter 2]

7 Oct 2021

Comments and edits contributed by Academic Editor and distinguished Reviewers were very helpful. We learned by doing. Your encouragement was great. The time given to us for editing and resubmission was adequate. The instructions given by the Academic Editor were very clear. Many thanks.

---

## [Editor Report · Decision Letter 3]

18 Oct 2021

Trends in non-pharmaceutical intervention (NPI) related community practice for the prevention of COVID-19 in Addis Ababa, Ethiopia

PONE-D-21-03031R3

Dear Dr. Tefera,

We’re pleased to inform you that your manuscript has been judged scientifically suitable for publication and will be formally accepted for publication once it meets all outstanding technical requirements.

There remain some minor typographical errors eg " Despite the p public enforcement," and double periods in places  and the references have case issues and the journal name abbreviations are not all in the correct format. Please work with the publisher to make the necessary corrections. 

Kind regards,

John Conly, MD

Academic Editor

PLOS ONE

---

## [Editor Report · Acceptance letter]

15 Nov 2021

PONE-D-21-03031R3 

Trends in non-pharmaceutical intervention (NPI) related community practice for the prevention of COVID-19 in Addis Ababa, Ethiopia 

Dear Dr. Kumie:

I'm pleased to inform you that your manuscript has been deemed suitable for publication in PLOS ONE. Congratulations! Your manuscript is now with our production department. 

Kind regards, 

on behalf of

Dr John Conly 

Academic Editor

PLOS ONE